# Synbols: Probing Learning Algorithms with Synthetic Datasets

**Alexandre Lacoste**[1], **Pau Rodríguez**[1], Frédéric Branchaud-Charron[1], Parmida Atighehchian[1], Massimo Caccia[1,2], Issam Laradji[1], Alexandre Drouin[1], Matt Craddock[1], Laurent Charlin[2], and David Vázquez[1]

[1]Element AI
{allac, pau.rodriguez, frederic.branchaud-charron, parmida, massimo.caccia, issam.laradji, adrouin, matt.craddock, dvazquez}@elementai.com
[2]Mila, Université de Montréal
{massimo.p.caccia, lcharlin}@gmail.com

## Abstract

Progress in the field of machine learning has been fueled by the introduction of benchmark datasets pushing the limits of existing algorithms. Enabling the design of datasets to test specific properties and failure modes of learning algorithms is thus a problem of high interest, as it has a direct impact on innovation in the field. In this sense, we introduce Synbols — Synthetic Symbols — a tool for rapidly generating new datasets with a rich composition of latent features rendered in low resolution images. Synbols leverages the large amount of symbols available in the Unicode standard and the wide range of artistic font provided by the open font community. Our tool's high-level interface provides a language for rapidly generating new distributions on the latent features, including various types of textures and occlusions. To showcase the versatility of Synbols, we use it to dissect the limitations and flaws in standard learning algorithms in various learning setups including supervised learning, active learning, out of distribution generalization, unsupervised representation learning, and object counting.

## 1  Introduction

Open access to new datasets has been a hallmark of machine learning progress. Perhaps the most iconic example is ImageNet [9], which spurred important improvements in a variety of convolutional architectures and training methods. However, obtaining state-of-the-art performances on ImageNet can take up to 2 weeks of training with a single GPU [51]. While it is beneficial to evaluate our methods on real-world large-scale datasets, relying on and requiring massive computation cycles is limiting and even contributes to biasing the problems and methods we develop:

- **Slow iteration cycles**: Waiting weeks for experimental results reduces our ability to explore and gather insights about our methods and data.

- **Low accessibility**: It creates disparities, especially for researchers and organizations with limited computation and hardware budgets.

- **Poor exploration**: Our research is biased toward fast methods.

- **Climate change impact**: Recent analyses [48, 28] conclude that the greenhouse gases emitted from training very large-scale models, such as transformers, can be equivalent to 10 years' worth of individual emissions.[1]

A common alternative is to use smaller-scale datasets, but their value to develop and debug powerful methods is limited. For example, image classification datasets such as MNIST [31], SVHN [38] or CIFAR [24] each contain less than 100,000 low-resolution ($32 \times 32$ pixels) images which enables short learning epochs. However, these datasets provide a single task and can prevent insightful model comparison since, e.g., modern learning models obtain above 99% accuracy on MNIST.

In addition to computational hurdles, *fixed datasets* limit our ability to explore non-i.i.d. learning paradigms including out-of-distribution generalization, continual learning and, causal inference. I.e., the algorithms can latch onto spurious correlations, leading to highly detrimental consequences when the evaluation set comes from a different distribution [3]. Similarly, learning disentangled representations requires non i.i.d. data for both training and properly evaluating [34, 20, 46]. This raises the need for good synthetic datasets with a wide range of latent features.

We introduce Synbols[2], an easy to use dataset generator with a rich composition of latent features for lower-resolution images. Synbols uses Pycairo, a 2D vector graphics library, to render UTF-8 symbols with a variety of fonts and patterns. Fig. 1 showcases generated examples from several attributes (§ 2 provides a complete discussion). To expose the versatility of Synbols, we probe the behavior of popular algorithms in various sub-fields of our community. Our contributions are:

- Synbols: a dataset generator with a rich latent feature space that is easy to extend and provides low resolution images for quick iteration times (§ 2).
- Experiments probing the behavior of popular learning algorithms in various machine-learning settings including: the robustness of supervised learning and unsupervised representation-learning approaches w.r.t. changes in latent-data attributes (§ 3.1 and 3.4) and to particular out-of-distribution patterns (§ 3.2), the efficacy of different strategies and uncertainty calibration in active learning (§ 3.3), and the effect of training losses for object counting (§ 3.5).

**Related Work**  Creating datasets for exploring particular aspects of methods is a common practice. Variants of MNIST are numerous [37, 41], and some such as colored MNIST [3, 2] enable proofs of concept, but they are fixed. dSprites [36] and 3D Shapes [21] offer simple variants of 2D and 3D objects in $64 \times 64$ resolution, but the latent space is still limited. To satisfy the need for a richer latent space, many works leverage 3D rendering engines to create interesting datasets (e.g, CLEVR [19], Synthia [44], CARLA [10]). This is closer to what we propose with Synbols, but their minimal viable resolution usually significantly exceeds $64 \times 64$, which requires large-scale computation.

## 2 Generator

The objective of the generator is to provide means of playing with a wide variety of concepts in the latent space while keeping the resolution of the image low. At the same time, we provide a high level interface that makes it easy to create new datasets with diverse properties.

### 2.1 Attributes

**Font Diversity**  (Fig. 1a) To obtain a variety of writing styles, we leverage the large quantity of artistic work invested in creating the pool of open source fonts available online. When creating a new font, an artist aims to achieve a style that is diversified while maintaining the readability of the underlying symbols in a fairly low resolution. This is perfectly suited for our objective. While many fonts are available under commercial license, fonts.google.com provides a repository of open source fonts.[3] This leaves us with more than 1,000 fonts available across a variety of languages. To

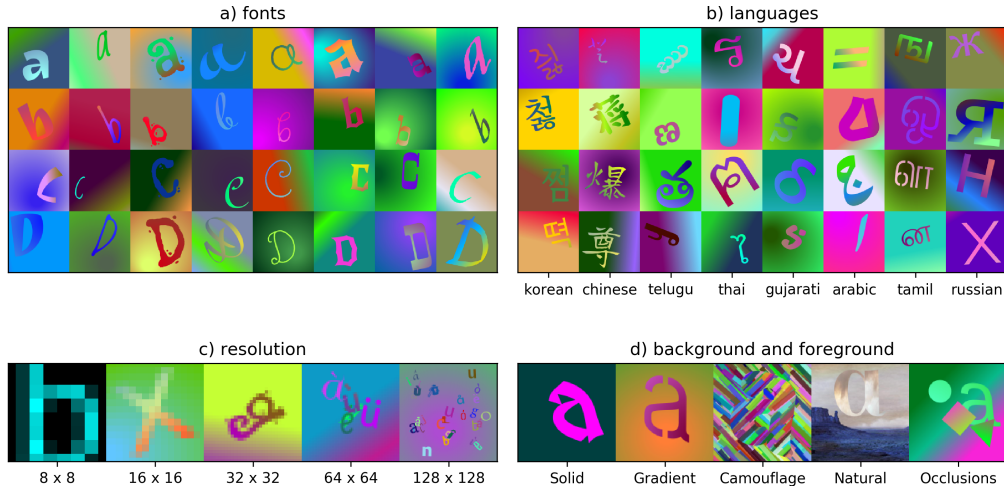

Figure 1: Generated symbols for different fonts, alphabets, resolutions, and appearances.

filter out the fonts that are almost identical to each other, we used the confusion matrix from a Wide ResNet [52] trained to distinguish all the fonts on a large version of the dataset.

**Different Languages**   (Fig. 1b) The collection of fonts available at fonts.google.com spans 28 languages. After filtering out languages with less than 10 fonts or rendering issues, we are left with 14 languages. Interestingly, the Korean alphabet (Hangul) contains 11,172 syllables, each of which is a specific symbol composed of 2 to 5 letters from an alphabet of 35 letters, arranged in a variable 2 dimensional structure. This can be used to test the ability of an algorithm to discriminate across a large amount of classes and to evaluate how it leverages the compositional structure of the symbol.

**Resolution**   (Fig. 1c) The number of pixels per image is an important trade-off when building a dataset. High resolution provides the opportunity to encode more concepts and provide a wider range of challenges for learning algorithms. However, it comes with a higher computational cost and slower iteration rate. In Fig. 1c-left, we show that we can use a resolution as low as $8 \times 8$ and still obtain readable characters.[4] Next, the $16 \times 16$ resolution is big enough to render most features of Synbols without affecting readability, provided that the range of scaling factors is not too large. Most experiments are conducted in $32 \times 32$ pixels, a resolution comparable to most small resolution datasets (e.g., MNIST, SVHN, CIFAR). Finally, a resolution of $64 \times 64$ is enough to generate useful datasets for segmentation, detection and counting.

**Background and Foreground**   (Fig. 1d) The texture of the background and foreground is flexible and can be defined as needed. The default behavior is to use a variety of shades (linear or radial with multiple colors). To explore the robustness of unsupervised representation learning, we have defined a *Camouflage* mode, where the color distribution of independent pixels within an image is the same for background and foreground. This leaves the change of texture as the only hint for detecting the symbol. It is also possible to use natural images as foreground and background. Finally, we provide a mechanism for adding a variety of occlusions or distractions using any UTF-8 symbol.

**Attributes**   Each symbol has a variety of attributes that affect the rendering. Most of them are exposed in a Python dict format for later usage such as a variety of supervised prediction tasks, as some hidden ground truth for evaluating the algorithm, or simply a way to split the dataset in a non i.i.d. fashion. The list of attributes includes *character*, *font*, *language*, *translation*, *scale*, *rotation*, *bold*, and *italic*. We also provide a segmentation mask for each symbol in the image.

## 2.2   Interface

The main purpose of the generator is to make it easy to define new datasets. Hence, we provide a high-level interface with default distributions for each attribute. Then, specific attributes can be redefined in Python as follows:

```
sampler = attribute_sampler(scale=0.5, translation=lambda: np.random.uniform(-2, 2, 2))
dataset = dataset_generator(sampler, n_samples=10000)
```

The first line fixes the scale of every symbol to 50% and redefines the distribution of the x and y translation to be uniform between -2 and 2, (instead of -1, 1). The second line constructs a Python generator for sampling 10,000 images. Because of the modified translation, the resulting dataset will have some symbols partially cropped by the image border. We will see in Sec. 3.3 how this can be use to study the brittleness of some active learning algorithms.

Using this approach, one can easily redefine any attributes independently with a new distribution. When distributions need to be defined jointly, for studying e.g. latent causal structures, we provide a slightly more flexible interface.

Datasets are stored in HDF5 or NumPy format and contain images, symbols masks, and all attributes. The default (train, valid, test) partition is also stored to help reproducibility. We also use a combination of Docker, Git versioning, and random seed fixing to make sure that the same dataset can be recreated even if it is not stored.

## 3 Experiments

To expose the versatility of the dataset generator, we explore the behavior of learning algorithms across a variety of machine-learning paradigms. For most experiments, we aim to find a setup under which some algorithms fail while others are more resilient. For a concise presentation of results, we break each subsection into *goal*, *methodology* and *discussion* with a short introduction to provide context to the experiment. All results are obtained using a (train, valid, test) partition of size ratio (60%, 20%, 20%). Adam [22] is used to train all models, and the learning rate is selected using a validation set. Average and standard deviation are reported over 3 runs with different random seeds. More details for reproducibility are included in App. **??** and the experiments code is available on GitHub[5].

### 3.1 Supervised Learning

While MNIST has been the hallmark of small scale image classification, it offers very minor challenges and it cannot showcase the strength of new learning algorithms. Other datasets such as SVHN offer more diversity but still lack discriminative power. Synbols provides a varying degree of challenges exposing the strengths and weaknesses of learning algorithms.

**Goal:** Showcase the various levels of challenges for Synbols datasets. Establish reference points of common baselines for future comparison.

**Methodology:** We generate the **Synbols Default** dataset by sampling a lower case English character with a font uniformly selected from a collection of 1120 fonts. The translation, scale, rotation, bold, and italic are selected to have high variance without affecting the readability of the symbol. We increase the difficulty level by applying the **Camouflage**, and **Natural Images** feature shown in Fig. 1d. The **Korean** dataset is obtained by sampling uniformly from the first 1000 symbols. Finally we explore font classification using the **Less Variations** dataset, which removes the bold and italic features, and reduces the variations of scale and rotation. See App. **??** for previews of the datasets.

**Backbones:** We compare 7 models of increasing complexity. Unless specified, all models end with global average pooling (GAP) and a linear layer. **MLP**: A three-layer MLP with hidden size 256 and leaky ReLU non-linearities (72k parameters). **Conv4 Flat**: A four-layer CNN with 64 channels per layer with flattening instead of GAP, as described by [47] (138k parameters). **Conv4 GAP**: A variant of Conv4 with GAP at the output (112k parameters). **Resnet-12**: A residual network with 12 layers [16, 39] (8M parameters). **WRN**: A wide residual network with 28 layers and a channel multiplier of 4 [52] (5.8M parameters). **Resnet12+** and **WRN+** were trained with data augmentation consisting of random rotations, translation, shear, scaling, and color jitter. More details in App. **??**.

**Discussion:** Table 1 shows the experiment results. The MNIST column exposes the lack of discriminative power of this dataset, where an MLP obtains 98.5% accuracy. SVHN offers a greater

challenge, but the default version of Synbols is even harder. To estimate the aleatoric noise of the data, we experiment on a larger dataset using 1 million samples. In App. **??**, we present an error analysis to further understand the dataset. On the more challenging versions of the dataset (i.e., Camouflage, Natural, Korean, Less Variations) the weaker baselines are often unable to perform. Accuracy on the Korean dataset are surprisingly high, this might be due to the lower diversity of font for this language. For font classification, where there is less variation, we see that data augmentation is highly effective. The total training time on datasets of size 100k is about 3 minutes for most models (including ResNet-12) on a Tesla V100 GPU. For WRN+ the training time goes up to 16 minutes, and about $10\times$ longer for datasets of size 1M.

| Dataset<br>Label Set<br>Dataset Size | MNIST<br>10 Digits<br>60k | SVHN<br>10 Digits<br>100k | Synbols Default<br>48 Symbols<br>100k | <br>48 Symbols<br>1M | Camouflage<br>48 Symbols<br>100k | Natural<br>48 Symbols<br>100k | Korean<br>1000 Symbols<br>100k | Less Variations<br>1120 Fonts<br>100k |
|---|---|---|---|---|---|---|---|---|
| **MLP** | 98.51 ±0.02 | 85.04 ±0.21 | 14.83 ±0.40 | 63.05 ±0.80 | 4.08 ±0.08 | 5.02 ±0.08 | 0.12 ±0.02 | 0.11 ±0.03 |
| **Conv-4 Flat** | 99.32 ±0.06 | 90.74 ±0.27 | 68.51 ±0.66 | 89.45 ±0.19 | 32.35 ±1.51 | 19.43 ±1.01 | 1.62 ±0.13 | 0.21 ±0.04 |
| **Conv-4 GAP** | 99.06 ±0.07 | 88.32 ±0.21 | 70.14 ±0.41 | 90.87 ±0.11 | 29.60 ±0.55 | 25.60 ±1.05 | 33.58 ±4.65 | 3.16 ±0.38 |
| **ResNet-12** | **99.70** ±0.05 | 96.38 ±0.03 | 95.43 ±0.12 | 98.85 ±0.02 | 90.14 ±0.05 | 81.21 ±0.46 | 97.08 ±0.13 | 39.41 ±0.30 |
| **ResNet-12+** | **99.73** ±0.05 | 97.19 ±0.04 | 97.16 ±0.05 | 99.44 ±0.00 | 94.09 ±0.07 | 85.80 ±0.15 | 98.54 ±0.07 | 57.42 ±0.50 |
| **WRN-28-4** | 99.64 ±0.06 | 96.07 ±0.07 | 93.57 ±0.29 | 98.88 ±0.04 | 86.34 ±0.16 | 73.26 ±0.53 | 95.79 ±0.51 | 23.10 ±0.90 |
| **WRN-28-4+** | **99.74** ±0.03 | **97.30** ±0.05 | **97.41** ±0.04 | **99.57** ±0.01 | **95.55** ±0.25 | **88.30** ±0.23 | **99.14** ±0.09 | **68.42** ±1.11 |

Table 1: **Supervised learning:** Accuracy of various models on supervised classification tasks. Results within 2 standard deviations of the highest accuracy are in **bold**.

## 3.2 Out of Distribution Generalization

The supervised learning paradigm usually assumes that the training and testing sets are sampled independently and from the same distribution (i.i.d.). In practice, an algorithm may be deployed in an environment very different from the training set and have no guarantee to behave well. To improve this, our community developed various types of inductive biases [53, 12], including architectures with built-in invariances [30], data augmentation [30, 25, 8], and more recently, robustness to spurious correlations [3, 1, 26]. However, the evaluation of such properties on out-of-distribution datasets is very sporadic. Here we propose to leverage Synbols to peek at some of those properties.

**Goal:** Evaluate the inductive bias of common learning algorithms by changing the latent factor distributions between the training, validation, and tests sets.

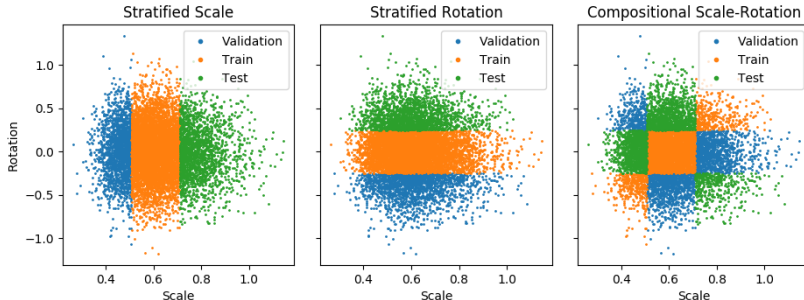

Figure 2: Example of the different types of split using scale and rotation

**Methodology:** We use the Synbols Default dataset (Sec. 3.1) and we partition the train, validation, and test sets using different latent factors. **Stratified** Partitions uses the first and last 20 percentiles of a continuous latent factor as the validation and test set respectively, leaving the remaining samples for training (Fig. 2). For discrete attributes, we randomly partition the different values. Solving such a challenge is only possible if there is a built-in invariance in the architecture. We also propose **Compositional** Partitions to evaluate the ability of an algorithm to compose different concepts. This is done by combining two stratified partitions such that there are large holes in the joint distribution while making sure that the marginals keep a comparable distribution (Fig. 2-right).

**Discussion:** Results from Table 2 show several drops in performance when changing the distribution of some latent factors compared to the i.i.d. reference point highlighted in blue. Most algorithms are barely affected in the **Stratified Font** column. This means that character recognition is robust to fonts

| Dataset Partition | i.i.d. | Synbols Default | | | | | Less Variations |
| --- | --- | --- | --- | --- | --- | --- | --- |
| | | Stratified Font | Stratified Scale | Stratified Rotation | Compositional Rot-Scale | Stratified x-Translation | Stratified Char |
| MLP | 14.83 ±.40 | **14.94** ±.37 | **15.59** ±.35 | 11.41 ±.05 | 7.91 ±.21 | 7.96 ±.06 | .08 ±.01 |
| Conv-4 Flat | 68.51 ±.66 | 67.17 ±.68 | **71.75** ±.17 | 5.54 ±.31 | 53.87 ±.89 | 44.78 ±.72 | .24 ±.01 |
| Conv-4 GAP | 70.14 ±.41 | **69.67** ±.18 | 62.85 ±.37 | 48.25 ±.38 | 54.88 ±.57 | 65.46 ±.37 | 2.97 ±.32 |
| Resnet-12 | 95.43 ±.12 | 94.62 ±.09 | 94.10 ±.13 | 82.37 ±.03 | 90.56 ±.20 | 94.62 ±.19 | 25.59 ±.22 |
| Resnet-12+ | 97.16 ±.05 | 96.10 ±.06 | 96.34 ±.07 | 91.96 ±.30 | 94.43 ±.26 | **96.84** ±.07 | 33.95 ±.61 |
| WRN-28-4 | 93.57 ±.29 | **93.27** ±.11 | 92.18 ±.16 | 79.53 ±.09 | 87.68 ±.46 | 92.99 ±.07 | 16.85 ±.23 |
| WRN-28-4+ | 97.41 ±.04 | 96.41 ±.08 | 96.80 ±.14 | 91.81 ±.48 | 95.16 ±.06 | **97.33** ±.16 | 35.41 ±2.39 |

Table 2: **Out of Distribution:** Results reporting accuracy on various train, valid, test partitions. Results in blue are the reference point for the Synbols Default dataset, results in red have a drop of more than 5% absolute accuracy compared to reference point, and **bold** results have a drop of less than 0.5% absolute accuracy compared to reference point. Refer to Table 1 for the reference point of font classification on the Less Variations dataset.

that are out of distribution.[6] On the other hand, all architectures are very sensitive to the **Stratified Rotation** partition, suffering from a systematic drop of at least 5% absolute accuracy compared to the i.i.d. evaluation. While data augmentation helps it still suffers from an important drop of accuracy. Scale is also a factor that affects the performance, but interestingly, some architectures are much more resilient than others.[7] Next, the **Compositional Rot-Scale** partition shows less performance drop than its stratified counterparts. This hints that neural networks are capable of some form of compositional learning. Moving on to the **Stratified x-Translation** partition, we see that many algorithms not affected by this intervention and retains its original performance. This is expected since convolution with global average pooling is invariant to translation. However, the MLP and Conv-4 Flat do not share this property and they are significantly affected. Finally, we observe an important drop of performance on font prediction when it is evaluated on characters that were not seen during training. This hints that many font classification results in Table 1 are memorizing pairs of symbols and font.

## 3.3 Active Learning

Instead of using a large supervised dataset, active learning algorithms aim to request labels for a small subset of unlabeled data. This is done using a query function seeking the most informative samples. A common query function is the predictive entropy (P-Entropy) [45]. However, a more principled method uses the mutual information between the model uncertainty and the prediction uncertainty of a given sample. This can be efficiently approximated using BALD [18]. However, in practice, it is common to observe comparable predictive performances between BALD and P-Entropy [5, 13]. We hypothesize that this may come from the lack of aleatoric uncertainty in common datasets. We thus explore a variety of noise sources, ranging from pixel noise to ablative noise in the latent space. We also explore the importance of uncertainty calibration using Kull et al. [27]. This is done by learning a Dirichlet distribution as an extra layer after the logits, while keeping the rest of the network fixed. To look at the optimistic scenario, we learn it using the full validation set with all the labels (after the network has converged using its labeled set).

**Goal:** Showcase the brittleness of P-Entropy on some types of noise. Search for cases where BALD may fail and see where calibrated uncertainty can help.

**Methodology:** We compare BALD and P-Entropy to random sampling. These methods are evaluated on different variations of Synbols Default. The **Label Noise** dataset introduces uniform noise in labels with probability 0.1. The **Pixel Noise** dataset adds $\epsilon \sim \mathcal{N}(0, 0.3)$ to each pixel in 10% of the images and clips back the pixel value to the range $(0, 1)$. The **10% missing** dataset omits the symbol with probability 0.1. In the **Cropped** dataset, translations are sampled uniformly between -2 and 2, yielding many symbols that are partly cropped. Finally, the **20% Occluded** dataset draws a large shape, occluding part of the symbol with probability 0.2. For concise results, we report the Negative Log Likelihood after labeling 10% of the unlabeled set. Performance at 5% and 20% labeling size can be found in Sec. **??**, along with implementation details and previews of all datasets.

| | CIFAR10 | No Noise | Label Noise | Pixel Noise | 10% Missing | Out of the Box | 20% Occluded |
|---|---|---|---|---|---|---|---|
| **BALD** | **0.59** ±0.03 | **0.85** ±0.09 | **2.05** ±0.06 | **0.91** ±0.02 | **1.24** ±0.04 | **1.96** ±0.03 | **1.39** ±0.05 |
| **P-Entropy** | **0.59** ±0.01 | **0.85** ±0.06 | **2.03** ±0.04 | 1.45 ±0.06 | 2.02 ±0.05 | 2.48 ±0.04 | 1.72 ±0.09 |
| **Random** | 0.66 ±0.04 | 0.98 ±0.05 | 2.12 ±0.11 | 1.00 ±0.04 | 1.32 ±0.07 | 2.04 ±0.04 | 1.51 ±0.04 |
| **BALD Calibrated** | **0.58** ±0.01 | **0.89** ±0.08 | **2.00** ±0.04 | **0.93** ±0.03 | **1.19** ±0.08 | 2.00 ±0.08 | **1.38** ±0.05 |
| **Entropy Calibrated** | **0.57** ±0.02 | **0.86** ±0.07 | **1.99** ±0.06 | 1.42 ±0.04 | 2.06 ±0.05 | 2.51 ±0.04 | 1.66 ±0.05 |

Table 3: **Active Learning** results reporting Negative Log Likelihood on test set after labeling 10% of the unlabeled training set. Results worse than random sampling are shown in red and results within 1 standard deviation of the best result are shown in **bold**.

**Discussion:** In Table 3, we recover the result where P-Entropy is comparable to BALD using CIFAR 10 and the No Noise dataset. Interestingly, the results highlighted in red show that P-Entropy will perform systematically worse than random when some images have unreadable symbols while BALD keeps its competitive advantage against Random. When training on 10% Missing, we found that P-Entropy selected 68% of it's queries from the set of images with omitted symbols vs 4% for BALD. We also found that label noise and pixel noise are not appropriate to highlight this failure mode. Finally, calibrating the uncertainty on a validation set did not significantly improve our results.

## 3.4 Unsupervised Representation Learning

Unsupervised representation learning leverages unlabeled images to find a semantically meaningful representation that can be used on a downstream task. For example, a Variational Auto-Encoder (**VAE**) [23] tries to find the most compressed representation sufficient to reconstruct the original image. In Kingma and Welling [23], the decoder is a deterministic function of the latent factors with independent additive noise on each pixel. This inductive bias is convenient for MNIST where there is a very small amount of latent factors controlling the image. However, we will see that simply adding texture can be highly detrimental on the usefulness of the latent representation. The Hierarchical VAE (**HVAE**) [54] is more flexible, encoding local patterns in the lower level representation and global patterns in the highest levels. Instead of reconstructing the original image, **Deep InfoMax** [17] optimizes the mutual information between the representation and the original image. As an additional inductive bias, the global representation is optimized to be predictive of each location of the image.

**Goal:** Evaluate the robustness of representation learning algorithms to change of background.

**Methodology:** We generate 3 variants of the dataset where only the foreground and background change. To provide a larger foreground surface, we keep the Bold property always active and vary the scale slightly around 70%. The other properties follow the Synbols Default dataset. The **Solid Pattern** dataset uses a black and white contrast. The **Shades** dataset keeps the smooth gradient from Default Synbols. The **Camouflage** dataset contains many lines of random color with the orientation depending on whether it is in the foreground or the background. Samples from the dataset can be seen in Sec. **??**. All algorithms are then trained on 100k samples from each dataset using a 64-dimensional representation. Finally, to evaluate the quality of the representation on downstream tasks, we fix the encoder and use the training set for learning an MLP to perform classification of the 26 characters or the 888 fonts. For comparison purposes, we use the same backbone as Deep InfoMax for our VAE implementation and we also explore using ResNet-12 for a higher capacity VAE.

| | Character Accuracy | | | Font Accuracy | | |
|---|---|---|---|---|---|---|
| | Solid Pattern | Shades | Camouflage | Solid Pattern | Shades | Camouflage |
| **Deep InfoMax** | 83.87 ±0.80 | 6.52 ±0.60 | 4.85 ±2.01 | 16.44 ±0.67 | 0.31 ±0.04 | **0.28** ±0.10 |
| **VAE** | 63.48 ±0.97 | 22.43 ±2.65 | 3.85 ±0.33 | 2.68 ±0.15 | 0.36 ±0.07 | 0.18 ±0.01 |
| **HVAE (2 level)** | 66.72 ±9.36 | 28.86 ±1.17 | 3.91 ±0.19 | 2.71 ±0.53 | 0.39 ±0.10 | 0.17 ±0.01 |
| **VAE ResNet** | 74.16 ±0.37 | 37.40 ±0.46 | 3.33 ±0.02 | 4.97 ±0.08 | 0.55 ±0.04 | 0.17 ±0.03 |
| **HVAE (2 level) ResNet** | 72.19 ±0.11 | **58.36** ±3.45 | 3.52 ±0.16 | 3.33 ±0.06 | **0.73** ±0.16 | 0.16 ±0.02 |

Table 4: **Unsupervised Representation Learning:** Results reporting classification accuracy on 2 downstream tasks. Highest performance is in **bold** and red exposes unexpected results.

**Discussion:** While most algorithms perform relatively well on the Solid Pattern dataset, we can observe a significant drop in performance by simply applying shades to the foreground and background. When using the Camouflage dataset, all algorithms are only marginally better than random predictions. Given the difficulty of the task, this result could be expected, but recall that in Table 1, adding camouflage hardly affected the results of supervised learning algorithms. Deep InfoMax

is often the highest performer, even against HVAE with a higher capacity backbone. However the performances on Camouflage are still very low with high variance indicating instabilities. Surprisingly, Deep InfoMax largely underperforms on the Shades dataset. This is likely due to the global structure of the gradient pattern, which competes with the symbol information in the limited size representation. This unexpected result offers the opportunity to further investigate the behavior of Deep InfoMax and potentially discover a higher performing variant.

## 3.5   Object Counting

Object counting is the task of counting the number of objects of interest in an image. The task might also include localization where the goal is to identify the locations of the objects in the image. This task has important real-life applications such as ecological surveys [4] and cell counting [7]. The datasets used for this task [14, 4] are often labeled with point-level annotations where a single pixel is labeled for each object. There exists two main approaches for object counting: detection-based and density-based methods. Density-based methods [32, 33] transform the point-level annotations into a density map using a Gaussian kernel. Then they are trained using a least-squares objective to predict the density map. Detection-based methods first detect the object locations in the images and then count the number of detected instances. LCFCN [29] uses a fully convolutional neural network and a detection-based loss function that encourages the model to output a single blob per object.

**Goal:** We compare between an FCN8 [35] that uses the LCFCN loss and one that uses the density loss function. The comparison is based on their sensitivity to scale variations, object overlapping, and the density of the objects in the images.

**Methodology:** We generate 5 datasets consisting of $128 \times 128$ images with a varying number of floating English letters on a shaded background. With probability 0.7, a letter is generated as 'a', otherwise, a letter is uniformly sampled from the remaining English letters. We use mean absolute error (MAE) and grid average mean absolute error (GAME) [15] to measure how well the methods can count and localize symbol 'a'. We first generate a dataset with symbols of **Fixed Scale** and of **Variable Scale** where the number of symbols in each image is uniformly sampled between 3 and 10. These two datasets are further separated into **Non-Overlapping** and **Overlapping** subsets where images are said to overlap when the masks of any two symbols have at least one pixel that coincides. The **Variable Scale** dataset is used to evaluate the sensitivity of the methods to scaling where scale $\sim 0.1e^{0.5\epsilon}$, and $\epsilon \sim \mathcal{N}(0,1)$, whereas the **Overlapping** dataset is used to evaluate how well the counting method performs under occlusions. The fifth dataset is **Crowded**, which evaluates the model's ability to count with a large density of objects in an image. It consists of images where the number of symbols, all with the same scale, is uniformly sampled between 30 and 50. Sample images from all datasets are shown in Sec. **??**.

**Discussion:** Table 5 reports the results of FCN8-D and FCN8-L, which are FCN8 trained with the density-based loss function [32] and the LCFCN loss function, respectively. For **Fixed Scale** and **Variable Scale**, the MAE and GAME results show that FCN8-L consistently outperforms FCN8-D when there is no overlapping. However, for setups with overlapping, FCN8-D's results are not affected much whereas the FCN8-L results degrade significantly. The reason is that separating overlapping objects is difficult when using a segmentation output instead a density map. On the other hand, FCN8-L has significantly better localization since it explicitly localizes the objects before counting them. For the **Crowded** dataset, the heavy occlusions make FCN8-L not train well at all. In this setup, density-based methods such as FCN8-D are clear winners for the MAE score as they do not require localizing individual objects before counting them. However, FCN8-L still outperforms FCN8-D in localization. This result suggests that density methods tend to output large density maps to get the count right as it is optimized for counting rather than localization.

| | Fixed scale | | Variable scale | | Crowded |
|---|---|---|---|---|---|
| | Non-overlapping | Overlapping | Non-overlapping | Overlapping | |
| **FCN8-L (MAE)** | **0.42** $\pm0.07$ | **1.06** $\pm0.09$ | **0.68** $\pm0.08$ | 1.52 $\pm0.44$ | <span style="color:red">7.50</span> $\pm2.89$ |
| **FCN8-D (MAE)** | 1.59 $\pm0.006$ | 1.20 $\pm0.35$ | 1.27 $\pm0.01$ | **1.40** $\pm0.01$ | **4.1** $\pm0.01$ |
| **FCN8-L (GAME)** | **0.53** $\pm0.04$ | **0.91** $\pm0.06$ | **0.59** $\pm0.04$ | **1.10** $\pm0.12$ | **6.69** $\pm0.67$ |
| **FCN8-D (GAME)** | 1.05 $\pm0.02$ | 1.49 $\pm0.57$ | 1.32 $\pm0.09$ | 2.09 $\pm0.04$ | <span style="color:red">8.14</span> $\pm0.42$ |

Table 5: **Object counting** results for exploring the sensitivity to scale variations, object overlapping and crowdedness. **Bold** indicates best results within 2 standard deviation and <span style="color:red">red</span> indicates failures.

### 3.6 Few Shot Learning

The large amount of classes available in Synbols makes it particularly suited for episodic meta-learning, where small tasks are generated using e.g., 5 classes and 5 samples per class. In this setting, a learning algorithm has to meta-learn to solve tasks with very few samples [40]. Interestingly, the different attributes of a symbol also provide the opportunity to generate different kinds of tasks and to evaluate the capability of an algorithm to adapt to new types of tasks.

Metric learning approaches [50, 47, 39, 42] usually dominate the leader board for few-shot image classification. They find a representation such that a similarity function can perform well at predicting classes. Perhaps the most popular is **ProtoNet** [47], where the representations of all images in a class are averaged to obtain the class prototype. Next, the closest prototype to a test image's representation is used for predicting its label. **RelationNet** [49] learns a shared similarity metric to compare supports and queries in an episode. Class probabilities are obtained with softmax normalization. For more flexible adaptation we also consider initialization-based meta-learning algorithms. **MAML** [11] does so by meta-learning the initialization of a network such that it can perform well on a task after a small number of gradient steps e.g., 10.

**Goal:** Evaluate the capability of meta-learning algorithms to adapt to new types of tasks.

**Methodology:** We generate a dataset spanning the 14 available languages. For a more balanced dataset, we limit it to 200 symbols and 200 fonts per language. This leads to a total of 1099 symbols and 678 fonts. Next, we evaluate our ability to solve font prediction tasks when meta-training on char tasks vs meta-training on font tasks. We also perform the converse to evaluate our ability to solve symbol prediction tasks at meta-test time. All tasks consist of 5 classes with 5 examples per classes, and the classes used to generate tasks during meta-test phase were never seen during the meta-train phase or meta-validation phase.

| Meta-Test | Characters | | Fonts | |
|---|---|---|---|---|
| Meta-Train | **Characters** | **Fonts** | **Fonts** | **Characters** |
| **ProtoNet** | 95.68 $\pm$0.42 | 75.73 $\pm$0.81 | 72.45 $\pm$1.52 | 43.13 $\pm$0.42 |
| **RelationNet** | 87.82 $\pm$2.22 | 57.00 $\pm$1.76 | 63.75 $\pm$4.82 | 38.81 $\pm$1.20 |
| **MAML** | 91.07 $\pm$0.69 | 66.40 $\pm$4.73 | 68.65 $\pm$1.87 | 40.68 $\pm$0.24 |

Table 6: **Few-Shot Learning:** Accuracy over 5 classes and 5 samples per class for different configurations of types of tasks.

**Discussion:** The first column of Table 6 exhibits the expected behavior of few-shot learning algorithms i.e., the accuracy is high and ProtoNet is leading the board. When looking at the other columns, we observe an important drop in accuracy when the Meta-Train set and the Meta-Test set don't share the same distribution of tasks.[8] This is to be expected, but the magnitude of the drop casts doubts over the re-usability of the current meta-learning algorithms. By highlighting this use-case, we hope to provide a way to measure progress in this direction. Our results are inline with the ones presented in recent work [6] which leverages the Synbols dataset to show that MAML trained on character classification tasks also incurs an important drop when tested on font classification tasks.

## 4 Conclusion

This work presented Synbols, a new tool for quickly generating datasets with a rich composition of latent features. We showed its versatility by reporting numerous findings across 5 different machine learning paradigms. Among these, we found that using prediction entropy for active learning can be highly detrimental if some images have insufficient information for predicting the label.

As future work, we intend to support video generation, and visual question answering. Other attributes such as symbol border, shadow, and texture can also be added to create even more precise and insightful diagnostics.

## Broader Impact

The introduction of benchmark new datasets has historically fueled progress in machine learning. However, recent large-scale datasets are immense, which makes ML research over-reliant on massive computation cycles. This biases research advances towards fast and computationally-intensive methods, leading to economic and environmental impacts. Economically, reliance on massive computation cycles creates disparities between researchers and organizations with limited computation and hardware budgets, versus those with more resources. With regard to the environment and climate change, recent analyses [48, 28] conclude that the greenhouse gases emitted from training very large-scale models, such as transformers, can be equivalent to 10 years' worth of individual emissions. While these impacts are not unique to ML research, they are reflective of systemic challenges that ML research could address by developing widely available, high-quality, and diverse datasets that mimic real-world concepts and are conscious of computational hurdles.

The Synbols synthetic dataset generator is designed to explore the behavior of learning algorithms and discover brittleness to certain configurations. It is expected to stimulate the improvement of core properties in vision algorithms:

- Identifiability of latent properties
- Reusability of machine learning models in new environments
- Robustness to changes in the data distribution
- Better performance on small datasets

We designed Synbols with diverse and flexible features (i.e., font diversity, different languages, lower resolution for POC, flexible texture of the background and foreground) and we demonstrated its versatility by reporting numerous findings across 5 different machine learning paradigms. Its characteristics address the economic and environmental challenges and we expect this tool to have a transversal impact on the field of machine vision with potential impact on the field of machine learning. Its broader impacts, both positive and negative, will be guided by the progress that it stimulates in the machine vision community, where potential applications range from autonomous weapons to climate change mitigation. Nevertheless, we hope that our work will help develop more robust and reliable machine learning algorithms while reducing the amount of greenhouse gas emissions from training by way of its smaller scale. Economic impact: reliance on computing-intensive environments creates disparities, especially for researchers and organizations with limited computation and hardware budgets. Environmental impact: Recent analyses [45, 28] conclude that the greenhouse gases emitted from training very large-scale models, such as transformers, can be equivalent to 10 years' worth of individual emissions

## Acknowledgments and Disclosure of Funding

This work was majorly supported by Element AI. Laurent Charlin is supported through a CIFAR AI Chair and grants from NSERC, CIFAR, IVADO, Samsung, and Google.

## Footnotes

[1]This analysis includes hyperparameter search and is based on the average United States energy mix, largerly composed of coal. The transition to renewable-energy can reduce these numbers by a factor of 50 [28]. Also, new developments in machine learning can help mitigate climate change [43] at a scale larger than the emissions coming from training learning algorithms.

[2]https://github.com/ElementAI/synbols

[3]Each font family is available under the Open Font License or Apache 2.

[4]For readability, scale and rotation are fixed and solid colors are used for the background and foreground

[5]https://github.com/ElementAI/synbols-benchmarks

[6]This is not too surprising since there is a large amount of fonts and many share similarities.

[7]The test partition uses larger scale which tend to be easier to classify, leading to higher accuracy.

[8]Note that for a given task, the train and the test sets comes from the same distribution

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
