[Supplementary Material]

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

# A  Emissions and Energy Usage

Experiments were conducted using private infrastructure, with a carbon efficiency of 0.026 $kgCO_2eq/kWh$. A cumulative total of 23916 hours of computation was performed on hardware of type Tesla V100 (TDP of 300W). This includes all phases of the project including debugging, failed experiments and hyperparameter search.

Total emissions are estimated to be 186.54 $kgCO_2eq$ of which 1000 $kgCO_2eq$ were compensated using Gold Standard credits.

Estimations were conducted using the MachineLearning Impact calculator presented in Lacoste et al. [34].

# B  Extended Details

For each experiment of Section 3, we provide more details. Implementation details are described for reproducibility purposes, and samples for each datasets are presented. Also, when available, we provide extended results.

## B.1  Supervised Learning

We provide implementation details for algorithms of Section 3.1 and present samples for each dataset in Figure 3. In addition, we present error analysis on the Default 1M dataset.

Figure 3: **Supervised Learning:**. Samples from datasets of Section 3.1. **Upper Left:** Default Symbol dataset. **Upper Center:** Camouflage dataset. **Upper Right** Natural Images dataset. **Lower left:** 1000 Korean Syllables dataset. **Lower Right:** Less Variations dataset.

### B.1.1  Implementation Details

For all experiments we trained a three-layer MLP, a 4-layer CNN with 64 channels per layer and max pooling [54], a Resnet-12 [20, 45], and a WRN-28-4 [59].

As specified in [45], Resnet-12 is composed of four residual blocks with three $3 \times 3$ convolutional layers per block and output size 64, 128, 256, and 512, respectively. All the layers are followed by

Figure 4: Error analysis on Synbols Default 1M with WRN-28-4+ architecture. **Left:** Confusion matrix with ground truth on the vertical axis and predictions on the horizontal axis. **Right:** Example of errors for the top 10 most confused letters.

batch normalization [24] and ReLU non-linearities. Dropout of 0.1 is placed after the first and second convolution of each block. Max pooling is performed at the output of each block.

WRN is a 28-layer residual architecture [20]. In this architecture, the input is first fed to a 16 channel $3\times3$ convolution, followed by three groups of four residual blocks with two convolutions per block. Blocks in the same group share the same amount of channels: 32, 64, and 128, respectively. WRN-28-4 uses $\times4$ that amount of channels. The last two groups start with a stride of 2 to reduce the dimensionality of the input. A dropout of 0.1 is placed after the first convolution of each block.

The data augmentation consists of uniformly sampled affine deformations. Concretely, rotations are sampled in $[-10, 10]$ degrees. Shear is sampled uniformly in $[-2, 2]$ degrees. Vertical and horizontal translations are sampled uniformly in $[-10\%, 10\%]$ pixels. Scaling is sampled in $[0.9, 1.1]\times$ original scale.

All the models are trained with Adam [28] and a learning rate of $4 * 10^{-4}$ for 200 epochs with a batch size of 512. For WRN, the batch size is 128 and the learning rate is $10^{-4}$. Early stopping is performed after 10 epochs without decrease in validation loss. Models are trained with mixed precision.[9]

### B.1.2 Error Analysis

In Figure 4, we report a confusion matrix and error analysis for WRN-28-4+ on Synbols Default 1M. Errors are as expected: 'i' and 'l' are the most confused characters, followed by 'v' which often looks like 'u' on some fonts. We can also observe that many errors comes from bogus fonts, where 2 of them render a rectangle instead of the symbol and another original font prints bar-codes with a very small version of the character. These 3 fonts are now removed from Synbols.

### B.2  Out of Distribution Generalization

We provide implementation details for algorithms of Section 3.1. The dataset are the same as in Section 3.1 and samples are presented in Figure 3.

### B.2.1 Implementation Details

### B.3 Active learning

We provide implementation details for algorithms of Section 3.3 and present samples for each dataset in Figure 5. In addition, we present extended results.

### B.3.1 Implementation Details

For all experiments, we train a VGG-16 [61] trained on Imagenet [11] until convergence on a validation set or a maximum of 10 epochs. Our active learning loop is implemented using the BaaL [5] library. We estimate epistemic uncertainty using MC-Dropout [16] using 20 samples. We also perform 20 Monte-Carlo sampling when computing the test performance. At each step, we label 100 images before resetting the weights to their original values as in Gal et al. [17]. For all reported results, we report the mean and standard deviation across 3 trials.

For calibration, we train the extra calibration layer for 10 epochs on the validation set. Following Kull et al. [33], we use the Adam optimizer and train two times, the latter with a reduced learning rate.

Figure 5: **Active Learning:**. Samples from datasets of Section 3.3. **Upper Left:** 50% pixel noise dataset. **Upper Right:** 10% Missing dataset. **Lower left:** Out of the Box dataset. **Lower Right:** 20% Occluded dataset.

### B.3.2 Extended Results

### B.4 Unsupervised Representation Learning

We provide implementation details for algorithms of Section 3.4 and present samples for each dataset in Figure 7. In addition, we present qualitative results.

Figure 6: Active learning convergence speed for various datasets. **Left:** Original dataset., **Center:** 10% Missing dataset. **Right:** 20% Occluded dataset.

| @5% | no-noise | label noise | pixel noise | 10% missing | out of the box | 20% occluded |
|---|---|---|---|---|---|---|
| **BALD** | **1.32** ±0.11 | **2.51** ±0.12 | **1.40** ±0.14 | 1.92 ±0.70 | **2.62** ±0.20 | **1.95** ±0.15 |
| **P-Entropy** | 1.47 ±0.05 | 2.66 ±0.14 | 2.33 ±0.11 | 2.56 ±0.22 | 2.97 ±0.15 | 2.48 ±0.11 |
| **Random** | 1.42 ±0.10 | 2.69 ±0.39 | **1.52** ±0.11 | 1.84 ±0.04 | 2.66 ±0.09 | 1.94 ±0.06 |
| **BALD calibrated** | **1.33** ±0.06 | 2.67 ±0.10 | **1.41** ±0.06 | 1.71 ±0.15 | 2.73 ±0.23 | 1.94 ±0.15 |
| **Entropy calibrated** | 1.46 ±0.05 | 2.77 ±0.17 | 2.33 ±0.26 | 2.50 ±0.08 | 3.03 ±0.14 | 2.58 ±0.28 |

| @20% | no-noise | label noise | pixel noise | 10% missing | out of the box | 20% occluded |
|---|---|---|---|---|---|---|
| **BALD** | 0.53 ±0.04 | **1.64** ±0.03 | **0.65** ±0.02 | **0.91** ±0.01 | 1.51 ±0.03 | 1.01 ±0.02 |
| **P-Entropy** | 0.49 ±0.02 | **1.65** ±0.02 | 0.69 ±0.04 | 0.93 ±0.02 | 1.67 ±0.02 | **0.96** ±0.03 |
| **Random** | 0.66 ±0.01 | 1.76 ±0.01 | 0.74 ±0.02 | 1.01 ±0.05 | 1.58 ±0.04 | 1.18 ±0.02 |
| **BALD calibrated** | 0.61 ±0.12 | **1.66** ±0.02 | **0.64** ±0.04 | **0.90** ±0.01 | 1.47 ±0.03 | **0.99** ±0.03 |
| **Entropy calibrated** | **0.48** ±0.04 | 1.76 ±0.02 | 0.69 ±0.02 | 0.99 ±0.00 | 1.69 ±0.02 | 0.98 ±0.03 |

Table 7: **Active Learning** results reporting Negative Log Likelihood on test set after labeling at 5% and 20% of the unlabeled training set. Results worse than random sampling are shown in red and results within 1 standard deviation of the best result are shown in **bold**.

### B.4.1 Implementation Details

For Deep Infomax [22], we use the pytorch implementation in `https://github.com/DuaneNielsen/DeepInfomaxPytorch`.

For the VAE and HVAE, we use a residual architecture based on BigGAN [7]. The encoder consists of three residual blocks of two convolutions followed by average pooling. Each block has 128, 128, and 256 channels respectively. The decoder follows an inverse structure, with nearest neighbor upsampling at the input of each block. For the HVAE, we follow the structure of VLAE [62]. That is, a latent code is sampled from the output of the first block of the encoder and a linear projection of it is concatenated to the input of the last block of the decoder. For fair comparison we also provide the VAE and HVAE with the same encoder described by Hjelm et al. [22]. The latent code size is 64. Models are optimized by minimizing the beta-VAE loss function [21]:

$$\mathcal{L} = |\mathbf{x} - \hat{\mathbf{x}}| + \beta D_{KL}(q(\mathbf{z}|\mathbf{x})||p(\mathbf{z})), \tag{1}$$

where $\mathbf{x}$ are input images, $\hat{\mathbf{x}}$ are the VAE reconstructions, and $\mathbf{z}$ are latent codes. We set $\beta = 0.01$ for all experiments, and we use cyclical annealing [14].

For attribute classification on the latent codes, we train a 3-layer MLP with hidden-size 256 on the training latent codes. We report classification and regression scores on the validation dataset.

### B.4.2 Qualitative results

In Figure 8, we provide reconstructions of the VAE and HVAE with the residual architecture. As can be seen there, the HVAE recovers sharper reconstructions. For the original dataset, the difference is more visible when the background and foreground share the same color (for instance, letter "K"). For the black and white dataset, the difference can be found in details, for instance, see the "I", or the

Figure 7: **Unsupervised Representation Learning:** Samples from datasets of Section 3.4 **Left:** Solid Pattern dataset. **Center:** Shades dataset. **Right:** Camouflage dataset.

bottom of "M". For camouflage, the VAE was unable to generate some of the characters, while they can still be recognized in the HVAE reconstructions, see letter "R".

## B.5 Object Counting

We provide implementation details in Section B.5.1 and present samples for the object counting datasets in Figure 9.

### B.5.1 Implementation Details

FCN-L and FCN-D use an Imagenet-pretrained VGG16 FCN8 network [41]. The models are trained with a batch size of 1 for 100 epochs with ADAM [28] and a learning rate of $10^{-4}$. The reported scores are on the test set which were obtained with early stopping on the validation set. The Gaussian sigma for FCN-D was chosen to be 1 and the weights of the coefficients for the 4 terms in the FCN-L loss were also chosen as 1.

## B.6 Few-shot Learning

For this few-shot learning benchmark, we perform 5-way classification. We fix the support and query size to 5 and 15, respectively.

### B.6.1 Implementation Details

For all experiments, we use a 4-layer convolutional neural network with 64 hidden units as commonly used in the few-shot literature [57, 54, 56]. All the methods are implemented using the PyTorch library [46], run on a single 12GB GPU and 8 CPUs .

For all methods, we use ADAM [28]. However, for MAML, in the inner-loop we use SGD. We use a batch size of one episode.

**Hyper-parameter search** For ProtoNet and RelationNet, we run a random search on the learning rates $\{0.005, 0.002, 0.001, 0.0005, 0.002, 0.0001\}$. We also search the patience hyper-parameter for early-stopping and for the learning rate schedule chosen in $\{10, 25, 50\}$ where the unit of measure is 500 training episodes. For MAML, we run random trials using the same learning rates as above, inner learning rates chosen from $\{0.5, 0.1, 0.01, 0.001, 0.0001\}$, and a number of inner updates chosen from $\{1, 2, 4, 8, 16\}$. Across the 4 settings presented in Table 6, we evaluate 14, 14, and 38 hyperparameter configurations for ProtoNet, RelationNet and MAML, respectively. For each configuration we repeat with 3 different random seeds. This results in a total of 34 days of compute to reproduce the current results.

Figure 8: Generation examples. **Left:** original, **center:** VAE reconstruction, **right:** HVAE reconstruction. Best viewed in color.

Figure 9: **Object Counting:** Samples from datasets of Section 3.5 **Upper Left:** Fixed Scale dataset. **Upper Right:** Varying Scale dataset. **Lower Center:** Crowded dataset.