[Reviews · NeurIPS 2020]

Review 1

Summary and Contributions: Paper presents “Synthetic Symbols (Synbols)”, a tool for generating images of unicode symbols with control over parameters like font, language, resolution, background texture, etc. Paper also presents experiments showing use of Synbols in standard supervised learning setting (establishing baselines), in out-of-distribution testing, for active learning, unsupervised representation learning, and object counting.

Strengths: - Paper is well-written. - There is a paradigm shift happening from datasets to dataset generators (e.g. simulators). This tool is aligned with that shift and might be broadly useful.

Weaknesses: It is difficult to characterize what new scientific understanding or knowledge was presented in this paper. Results are sections 3.1 and 3.2 appear entirely unsurprising (though admittedly, this could be hindsight bias). Results in 3.3 and 3.4 appear to present interesting insights (e.g. that P-Entropy and BALD appear to perform similar under no noise, but P-Entropy performs worse in the noisy setting, thus providing evidence for the hypothesis in L200). However, one is left wondering whether this insight generalizes beyond the specifics of this experiments/dataset (e.g. the way symbols were occluded) or whether we are creating an isolated MNIST++ sub-community. Of course, this is a general concern any time a diagnostic dataset/tool is introduced. Rarely is an immediately surprising insight offered in the same paper. The value of such datasets/tools is often clear only in hindsight with the benefit of time. If they are useful, they see organic adoption. If they are not, they organically fall to the wayside. Thus, even though I personally don’t see any particularly novel insight in this paper, the methodology followed is not wrong. And I happy to let the noisy process of science figure out the value here.

Correctness: Empirical claims appear well supported.

Clarity: Yes.

Relation to Prior Work: Largely yes. A few minor comments: - L58-59: Why? I don’t understand why you can’t render (and use) lower than 64x64 images from say CLEVR? - L261-263: I don't understand -- can't any object detection or instance segmentation datasets be used for counting? e.g. https://arxiv.org/abs/1903.02494 https://arxiv.org/abs/1604.03505

Reproducibility: Yes

Additional Feedback:


Review 2

Summary and Contributions: Authors introduce a tool for generating smaller-scale synthetic (low-resolution UTF8 symbols) datasets with a rich composition of latent features for debugging learning algorithms. They also provide a language-based interface for its generation. Generated synthetic datasets are evaluated to analyse the limitations/flaws in a number of learning tasks (e.g., supervised learning, active learning, out of distribution generalization, unsupervised representation learning, and object counting). Smaller-scale evaluation settings as the one proposed in the paper allow for quick iteration times.

Strengths: Comprehensive experimental setting. Illustrative sound results showing the usefulness of the tool for discriminating between learning algorithms. Easiness of use.

Weaknesses: Authors use a very limited number of learning nnet models used for evaluation: differences between those more proficient models are similar for some of the tasks addressed. Is it possible to “scale” the generation tool to address (future) more complex learning tasks, including further latent features for diversity, discrimination, or difficulty? As we have seen over the years, whenever a new (more difficult) AI benchmark appears on scene, researchers put all their effort on solving it, usually making their system very specialised. As pointed by the authors, the key issue when evaluating learning algorithms is to consider a greater diversity of problems. However, further analysis to determine what instances (images) are more difficult/discriminant (e.g. performing IRT analysis or just more simple studies taking into account whether difficulty is related to the size of the working memory, the size of the pattern, or the number of elements that need to be combined or retrieved from background knowledge) could also be of interest. These notions of difficulty are much more general and can work independently of the problem and the representation, although this is not very related to how challenging the problem is.

Correctness: The experimental setting follows the standard ML evaluation methodology.

Clarity: Very well written and structured.

Relation to Prior Work: Somehow limited and not very well introduced/connected.

Reproducibility: Yes

Additional Feedback: Looking at the other comments and the feedback provided by the authors (which have answered all the questions I raised), I have an even more positive feeling about the contributions of the paper.


Review 3

Summary and Contributions: The paper proposed a new data augmentation method to generate low resolution digit/text images with rich composition of latent features. The generated harder small dataset can be used for fast evaluation of new training algorithms.

Strengths: The task of constructing harder and non-fixed datasets for training and evaluation is of great practical important.

Weaknesses: 1. The proposed method seems only works for digit or text images, such as MNIST and SVHN. Can it be used on natural images, such as CIFAR10, which has wider applications in the real world then digit/text. 2. Are the results obtained on Synbols dataset generalizable to large-scale datasets? For example, if you find algorithm A is better than B on Synbols dataset, will the conclusion hold on large images (e.g., ImageNet scale) in real-world applications? This need to be discussed in the paper.

Correctness: I find no flaws in the proposed method.

Clarity: The paper is well written and easy to follow.

Relation to Prior Work: The relation to prior data augmentation works [1,2,3,4,5] (and many others in their references) is not well discussed. These data augmentation methods can be used for similar purpose of the proposed methods: to provide a harder training and evaluation benchmark with small images. These augmentation methods can be used on digit, text or natural images, which seems to have wider application scenarios than the proposed method. So what is the advantage of the proposed Synbols over these methods? [1] Cubuk, Ekin D., Barret Zoph, Dandelion Mane, Vijay Vasudevan, and Quoc V. Le. "Autoaugment: Learning augmentation strategies from data." In Proceedings of the IEEE conference on computer vision and pattern recognition, pp. 113-123. 2019. [2] Cubuk, Ekin D., Barret Zoph, Jonathon Shlens, and Quoc V. Le. "Randaugment: Practical automated data augmentation with a reduced search space." In Proceedings of the IEEE/CVF Conference on Computer Vision and Pattern Recognition Workshops, pp. 702-703. 2020. [3] Zhang, Xinyu, Qiang Wang, Jian Zhang, and Zhao Zhong. "Adversarial autoaugment." arXiv preprint arXiv:1912.11188 (2019). [4] Hendrycks, Dan, Norman Mu, Ekin D. Cubuk, Barret Zoph, Justin Gilmer, and Balaji Lakshminarayanan. "Augmix: A simple data processing method to improve robustness and uncertainty." arXiv preprint arXiv:1912.02781 (2019).

Reproducibility: Yes

Additional Feedback: Update after rebuttal: Thanks the authors for posting their valuable feedbacks. I acknowledge the difference between the proposed method and data augmentation. My comments about data augmentation are in the "Relation to prior work" section and only serves as a suggestion for the authors to add more discussions about previous work, not affecting my score. My major concern about this paper is that the proposed mothed and experiments are all for UTF-8 symbols instead of natural images, and I wonder whether the method and conclusions can be generalized to natural images. For example, for natural images, it can be non-trivial to do the foreground-background disentanglement. (Should they seek the time-consuming human annotation or use automatic object segmentation models with noisy background mask outputs?) It is important to validate the conclusions learned from UTF-8 datasets are generalizable to large-scale natural images, since one important motivation for debugging on small-scale UTF-8 datasets is to provide insights generalizable to large-scale real-world problems, as claimed by the authors in lines 19-21 and 30-31. That said, after rethinking, I would like to increase my score to 5 (marginally below the acceptance threshold), since the authors said they will generalize their method to natural images and more other applications in their “Future Work” section.


Review 4

Summary and Contributions: After the rebuttal, my score of the paper remains unchanged. I feel now more confident in the good contribution of the paper. ------------------ The trend of recent public datasets is towards the use of images with higher resolution and huge amount of images. Relying only with these for developing and evaluating novel methods is expensive because it requires huge resources which are only available to a part of the community, have slow iteration cycles and has an impact on the environment. The paper is about a novel tool for generating synthetic datasets which have certain features but are smaller in size and are generable on demand. A framework which exploits open fonts and open source libraries is proposed to automatically generate synthetic dataset, where resolution, diversity and attributes are all controllable. A large exploration with experiments on 5 different learning paradigm show that the generated datasets can be used in supervised learning, out of distribution predictions, active learning, unsupervised representation learning and counting. Quirks and brittleness are shown of several well known algorithms in their respective paradigm.

Strengths: + The paper is well presented, easy to read. + The experiments validates the generated datasets by the framework. They seem solid and cover 5 different ML paradigms. + The proposed tool can have a good impact on the community and help standardize several experiments with synthetic data. I was impressed by the versatility of the framework.

Weaknesses: - The attributes that affect the rendering may induce bias in the kind of images generated. There is no explicit discussion on the impact of such attributes. It would be interesting to discuss why and which attributes were selected and the impact on the bias of the generated datasets.

Correctness: The claims and the method seems to be solid and well executed.

Clarity: The paper is well written and easy to read.

Relation to Prior Work: Previous work are briefly discussed but include main related contributions of synthetic datasets. In fact, the use of synthetic dataset is a common practice in many papers, but few addresses it directly.

Reproducibility: Yes

Additional Feedback: - L108, "use" -> "used" - In table 2, all scores in the last column should be red since they have all a drop of 5% compared to column 1.

[Author Response · NeurIPS 2020]

**R1**, **R2**, **R3**, **R4**: We thank the reviewers for the numerous positive comments. **R4**: ''**The proposed tool can have a good impact on the community and help standardize several experiments with synthetic data. I was impressed by the versatility of the framework''**. **R3**: **"The task of constructing harder and non-fixed datasets for training and evaluation is of great practical important."**. **R1**: **"There is a paradigm shift happening from datasets to dataset generators (e.g. simulators). This tool is aligned with that shift and might be broadly useful.".** **R2**: **"Very well written and structured.'**

**R1**: **"one is left wondering whether this insight generalizes beyond the specifics of this experiments/dataset?"**
**R3**: **"Are the results obtained on Synbols dataset generalizable to large-scale datasets?"**
In the general case, one should always be careful on how scientific findings can generalize to other setups. We made sure to communicate this properly in the paper and encourage a good experiment design and, when possible, to test on real world datasets. Specifically, when a failure case is detected with Synbols, it is expected to hold in at least some real world scenarios. For example, we expect that P-Entropy would fail in real world scenarios when some objects are occluded in the dataset. Next, an algorithm that solves the problem in various simulation setup is expected to at least offer insights on real world solutions.

**R1**: **"It is difficult to characterize what new scientific understanding or knowledge was presented in this paper."**
We agree, many of the presented results are part of the wisdom of the more experimented researchers. While the aim of the paper is to provide a new tool for generating datasets (part of NeurIPS's CFP), we seized the opportunity to solidify and quantify this knowledge, which would otherwise remain intuitions.

**R1**: **"The value of such tools is often clear only in hindsight... If they are useful, they see organic adoption."**
For this purpose, we are investing effort on a high quality github repository with good documentation and ease of use. We will also properly advertise the tool to make sure it reaches the potential users.

**R1**: **"why you can't render (and use) lower than 64x64 images from say CLEVR?"**
In CLEVR multiple objects of different sizes are present in a single scene. When resizing the image, some of these objects become so small that they are reduced to a single pixel.

**R2**: **"Authors use a very limited number of learning nnet models used for evaluation"**
There is a total of 14 different backbones implemented across all the experiments. We are happy to add results from other backbones such as Squeeze-and-Excite Networks or other recommendations. The main limitation is the readability of the tables and the space available for describing the backbones, but we are happy to add many more in appendix, such as our few-shot learning experiments.

**R2**: **"differences between those more proficient models are similar for some of the tasks addressed. Is it possible to "scale" the generation tool to address (future) more complex learning tasks"**
Yes, the generator is prepared to be extended with harder benchmarks such as video generation, and VQA. Other attributes such as symbol border, shadow, and texture are also planned (see conclusion). Note that the font classification task already provides a challenging benchmark for current learning algorithms. Results in the last column of Table 1 report 80.35% vs 67.20% for the two best performing models.

**R2**: **"further analysis to determine what instances (images) are more difficult/discriminant (e.g. performing IRT analysis ..."**
This would make a really interesting experiment. We will add this for the camera ready. In addition, we will investigate how the overall difficulty of the dataset is affected by those attributes with complexity measures (Ho and Basu 2002).

**R3**: **"The paper proposed a new data augmentation method to generate low resolution digit/text images with rich composition of latent features."** **R3**: **"The proposed method seems only works for digit or text images, such as MNIST and SVHN. Can it be used on natural images, such as CIFAR10"**
Our work is not a tool for data augmentation. It is a tool for discovering model biases and it is not designed to generate natural images.

**R3**: **"The relation to prior data augmentation works [1-4] (and many others in their references) is not well discussed"** These data augmentation techniques are really interesting but orthogonal to our work. We now cite them in the related work section with a description of the differences between data augmentation and the proposed generator.

**R4**: **"It would be interesting to discuss why and which attributes were selected and the impact on the bias of the generated datasets."** In the methodology sub-sections of each experiments, we describe the distribution of each attributes and the intent of this choice. We also show samples of each generated dataset in Appendix to qualitatively verify the biases. We also updated the supplementary material to extend the discussion on this bias as requested.

**R4**: **"In table 2, all scores in the last column should be red since they have all a drop of 5% compared to col. 1."**
No, for font classification, the reference point is last column of Table 1. We made it more clear in the paper.

[Meta-Review · NeurIPS 2020]

This paper proposes a tool (Synbols) for generating datasets based on (augmented) images from unicode symbols sourced from thousands of fonts. The motivation behind this work is that recent trends on using large, high resolution images as datasets for improving and evaluating research methods requires a huge amount of compute that may not be available to everyone, and also have slow iteration cycles not to mention the energy used for training. The tool proposed allows the researcher to create smaller-scale synthetic datasets with control over parameters like font, language, resolution, textures, that can help facilitate the debugging, iteration, and development of new methods. They demonstrate the usefulness of their dataset in a number of tasks (supervised learning, active learning, out of distribution generalization, representation learning, objective counting), and demonstrate that their dataset can already be used to clearly identify limitations and flaws of existing well known algorithms in their respective paradigms. The reviewers have raised concerns and issues that were partially addressed in the authors’ response, which satisfied some of the reviewers. During the review process and discussion, we also clarified that this work is not about proposing a new data augmentation method, which all reviewers agreed on and the final evaluations and reviews are based on this assumption (although this can be made more clear in the writing's narrative IMO to avoid confusion). One weakness of the work is that the datasets may lack the sophistication of natural images which may limit its application, although I think as a tool for iterating new algorithms it is fine. After the review process and discussion, I agree with a few reviewers and feel, like R4, ""more confident in the good contribution of the paper"" and I think this will be a fine addition to the NeurIPS community. I am going to recommend acceptance (Poster), and I hope that the tool will be made readily available for the community to use later on, after the work is published.